# The genetic architecture of host response reveals the importance of arbuscular mycorrhizae to maize cultivation

M Rosario Ramírez-Flores[1†], Sergio Perez-Limon[2†], Meng Li[3], Benjamín Barrales-Gamez[2], Doris Albinsky[4], Uta Paszkowski[4], Víctor Olalde-Portugal[1], Ruairidh JH Sawers[3]*

[1]Departamento de Biotecnología y Bioquímica, Centro de Investigación y de Estudios Avanzados del Instituto Politécnico Nacional (CINVESTAV-IPN), Irapuato, Mexico; [2]Laboratorio Nacional de Genómica para la Biodiversidad/Unidad de Genómica Avanzada, Centro de Investigación y de Estudios Avanzados, Instituto Politécnico Nacional (CINVESTAV-IPN), Irapuato, Mexico; [3]Department of Plant Science, The Pennsylvania State University, State College, United States; [4]Crop Science Centre and Department of Plant Sciences, University of Cambridge, Cambridge, United Kingdom

**Abstract** Arbuscular mycorrhizal fungi (AMF) are ubiquitous in cultivated soils, forming symbiotic relationships with the roots of major crop species. Studies in controlled conditions have demonstrated the potential of AMF to enhance the growth of host plants. However, it is difficult to estimate the actual benefit in the field, not least because of the lack of suitable AMF-free controls. Here we implement a novel strategy using the selective incorporation of AMF-resistance into a genetic mapping population to evaluate maize response to AMF. We found AMF to account for about one-third of the grain production in a medium input field, as well as to affect the relative performance of different plant genotypes. Characterization of the genetic architecture of the host response indicated a trade-off between mycorrhizal dependence and benefit. We identified several QTL linked to host benefit, supporting the feasibility of breeding crops to maximize profit from symbiosis with AMF.

*For correspondence:
rjs6686@psu.edu

[†]These authors contributed equally to this work

Competing interests: The authors declare that no competing interests exist.

## Introduction

There is a pressing need to develop sustainable agricultural systems that improve productivity whilst minimizing adverse environmental impacts (*Lynch, 2019*). In the face of this challenge, there is great interest in the potential contribution of arbuscular mycorrhizal (AM) symbiosis and other microbial mutualisms (*Sawers et al., 2008*; *Fester and Sawers, 2011*; *Bender et al., 2016*). AM fungi (AMF) occur broadly in cultivated soils, forming symbiotic relationships with the roots of major crop species and, indeed, many commercial 'biofertilizers' are available on the market (*Faye et al., 2013*). Nonetheless, the value of AM symbiosis in agriculture remains a matter of debate (*Ryan and Graham, 2018*; *Rillig et al., 2019*). Experimental studies under controlled conditions have demonstrated that AMF have much to offer host plants, including enhanced uptake of immobile soil nutrients (especially phosphorus) and water, and improved tolerance to a range of abiotic and biotic stresses (*Borowicz, 2001*; *Chandrasekaran et al., 2014*; *Augé et al., 2015*; *Chiu and Paszkowski, 2019*). AMF also contribute important ecosystem services (*van der Heijden et al., 2015*), including promoting soil aggregation (*Rillig and Mummey, 2006*) and reducing nutrient loss through leaching and emission (*Bender et al., 2014*; *Cavagnaro et al., 2015*). However, efforts to integrate AMF into agricultural practice are hampered by the difficulty of evaluating host response in the field and the lack of

appropriate AMF free controls. When exogenous fungal inoculum is applied to a field, it is difficult to gauge its effectiveness with respect to the native fungal community; when looking to evaluate the impact of the native AMF community under any given agronomic scenario, there is no easy way to estimate what the baseline performance of plants would be if AMF were absent. Although experimental designs using chemical treatments and/or mechanical barriers can be implemented at a small scale, they are not practical for use in larger genetic studies or trials. To address these difficulties, we propose the incorporation of plant mycorrhizal mutants in the design of experiments to study AM symbiosis in the field.

AM symbiosis dates back over 450 million years (*Strullu-Derrien et al., 2018*). By contrast, agriculture is a recent development. Modern breeding efforts have focused on maximizing yield under homogeneous environments with large quantities of synthetic inputs, and it is not clear how the breeding process has impacted the AM symbiosis (discussed in *Sawers et al., 2008*; *Sawers et al., 2018*). Modern crop varieties retain the molecular machinery necessary for symbiotic establishment and are colonized to high levels in controlled experiments (*e.g. Gutjahr et al., 2008*). Elegant molecular genetic analyses, largely conducted in *Medicago truncatula* and *Lotus japonicus,* have identified a series of plant-encoded receptor and signal-transduction components necessary for AM symbiosis and, in relevant host plants, for interaction with nodule-forming rhizobia. Fungal signals are perceived by a partially-characterized suite of host LysM receptor kinases, acting in combination with the SYMRK (DMI2) receptor. Signal perception triggers cellular calcium spiking events, through the action of the nuclear ion-channels CASTOR and POLLUX (DMI1), resulting in the activation of downstream genes via the action of the CCaMK (DMI3) and CYCLOPS (IPD3) proteins (*Parniske, 2008*; *Zipfel and Oldroyd, 2017*). Notwithstanding the great antiquity of AM symbiosis, there is little genetic redundancy in this common symbiotic signaling pathway (CSSP) and single host gene mutations are sufficient to block symbiotic interaction (*e.g. Charpentier et al., 2008*; *Gutjahr et al., 2008*). Among the major cereal crops, CSSP mutations have only been reported in rice (*Charpentier et al., 2008*; *Gutjahr et al., 2008*), although the genes themselves can be unambiguously identified in the genomes of maize, wheat, and sorghum, amongst others (*Chiu and Paszkowski, 2020*).

The CSSP is essential for plant interaction with AMF but does not appear to be a major point of regulation of AM symbiosis with respect to either host development or environment. There are, however, many reports of intra-specific differences in the impact AM symbiosis has on the plant host (*e.g. Kaeppler et al., 2000*; *Mascher et al., 2014*; *Diedhiou et al., 2016*; *Sawers et al., 2017*; *Watts-Williams et al., 2019*; *Pawlowski et al., 2020*). In the case of rice, a variant of the OsCERK1 LysM receptor kinase present in the wild progenitor species *Oryza rufipogon* has been linked to greater colonization by AMF and enhanced host response, although this variant was not found to be segregating in the agronomically important *japonica* group (*Huang et al., 2020*). Apart from this interesting result, the mechanistic basis of host response variation remains largely uncharacterized. Typically, host response to AMF is defined as the difference between mycorrhizal and non-mycorrhizal plants, expressed in either absolute or relative terms (*Janos, 2007*). We further differentiate *dependence*, the capacity of a given variety to perform in the absence of AMF, and *benefit*, the degree to which a plant host profits from the association (discussed in *Janos, 2007*; *Sawers et al., 2010*). Importantly, variation in host response confounds differences in both dependence and benefit, with benefit being of the greater agronomic interest (*Hetrick et al., 1992*; *Diedhiou et al., 2016*; *Sawers et al., 2017*).

In this study we evaluate the impact of AMF in a cultivated field and distinguish dependence and benefit using the concepts of genotype x environment interaction (GEI; *Des Marais and Juenger, 2010*), considering mycorrhizal and non-mycorrhizal growth as two distinct 'environments'. To establish a non-mycorrhizal condition in the field, we selectively incorporated a mutation in the CSSP gene *Castor* into a bi-parental maize population for quantitative trait locus (QTL) mapping. The resulting population contained both AMF susceptible (*AMF-S*) and resistant (*AMF-R*) families (we use resistance here to denote an inability to from AM symbiosis due to host incompatibility), allowing us to map the interaction between plant trait QTL and AM symbiosis. We report a significant contribution of AM symbiosis in our field experiment, along with an empirical demonstration of dependence and benefit variation, and evidence of a trade-off between the two. The identification of benefit QTL supports the feasibility of breeding crops to maximize their advantage from AM symbiosis.

## Results

### Mutation in a maize common symbiosis gene demonstrates the importance of arbuscular mycorrhizae in the field

To generate AMF-resistant maize varieties for use in field experiments, we identified maize (var. B73) orthologs of the common-symbiosis genes *CASTOR* and *POLLUX* (*DMI1*) (*Parniske, 2008*; www.maizegdb.org). As in rice, *CASTOR* and *POLLUX* were found to be single copy genes in maize, and designated *ZmCastor* (GRMZM2G099160; Zm00001d012863; chromosome 5, 1 Mb) and *ZmPollux* (GRMZM2G110897; Zm00001d042694; chromosome 3, 177 Mb). We searched publicly available genetic resources (*McCarty et al., 2005*) and identified two *Mutator* (*Mu*) transposon insertions in the *Castor* gene, *castor-1* (44 bp upstream of the translational start site) and *castor-2* (39 bp downstream of the translational start site. *Figure 1A*). Following controlled inoculation with AMF, we saw fungal colonization of *castor-1* plants but found *castor-2* plants to be free from root-internal fungal structures, reflecting the different sites of transposon insertion in the two alleles (*Figure 1B*). We advanced *castor-2* (hereafter, *castor*) mutants for more detailed quantitative characterization, finding the transcript to be disrupted in the mutants and confirming the absence of fungal colonization following inoculation with either crude (sand-pot produced) or plate-cultured (in vitro produced) inoculum (*Figure 1C,D*).

Having identified a source of resistance to AMF, we proceeded to generate a population for mapping host response in the field. We crossed the *castor* mutant in its original temperate W22 background to the subtropical inbred line CML312 and performed two rounds of self-pollination to generate $F_{2:3}$ families. We used PCR genotyping to select $F_2$ individuals homozygous for either the wild-type or mutant allele at the *Castor* locus, such that the resulting $F_3$ families were 'fixed' with respect to the mutation. In this way, we generated a mapping population of 73 homozygous wild-type (susceptible; *AMF-S*) and 64 homozygous *castor* (resistant; *AMF-R*) $F_{2:3}$ families (*Figure 1—figure supplement 1*). Outside of the *Castor* locus, the different families segregated for W22 and CML312 gene content. The $F_2$ parents were further genotyped using genome-wide markers to build a genetic map for use in QTL mapping. We confirmed that resistance to AMF was stable in the field (*Figure 1—figure supplement 2*) and that under high-nutrient field conditions the *castor* mutation did not result in any gross defects in plant development or show marked pleiotropic effects beyond the block in mycorrhizal colonization (*Figure 1—figure supplement 3*). We note that our observations do not exclude more subtle, or environmentally dependent, pleiotropy.

To characterize the genetic architecture of host response to AMF in the field, we evaluated the 73 *AMF-S* and 64 *AMF-R* families in a replicated trial under rain-fed, medium-input, conditions (Ameca, Mexico. *Figure 2A*, *Figure 2—figure supplements 1*, *2* and *3*; three replicates, using three-row plots at 15 individuals per row). We chose an agronomic scenario that was representative of medium scale, subtropical production, and conditions under which we might expect AMF to be important. Overall, *AMF-R* families tended to exhibit mild-chlorosis (*Figure 2B*) and produce visibly poorer ears than *AMF-S* families (*Figure 2C–D*). We inspected the roots of a random sampling of plants, finding clear colonization by AMF of *AMF-S* families and an absence of AM fungal structures in *AMF-R* families. To estimate the overall host response, we compared all susceptible to all resistant families. We collected fifteen phenological and morphological traits and yield-components (*Figure 2E–H*, *Figure 2—figure supplement 4*; *Supplementary file 1*). Ten of fifteen traits differed significantly between *AMF-S* and *AMF-R* families, corresponding to positive host responses ranging from 3% to 51% (*Table 1*; positive defined here as greater values of growth traits and yield components or accelerated/more synchronous flowering). *AMF-R* families were reduced in stature and delayed in silking (female flowering), resulting in an extension of anthesis (male flowering)-silking interval (ASI; *Figure 2E,F*; *Supplementary file 1*) - a classic symptom of abiotic stress in maize. Ear size was reduced in *AMF-R* families, along with ear and total kernel weight (*Table 1*; *Figure 2—figure supplements 4* and *5*). Individual kernels were not significantly different in weight or size between *AMF-S* and *AMF-R* families (*Figure 2G*; *Table 1*). However, the total number of kernels per ear was reduced in *AMF-R* families (*Figure 2H*; *Table 1*), indicative of poor seed set, a possible consequence of increased ASI.

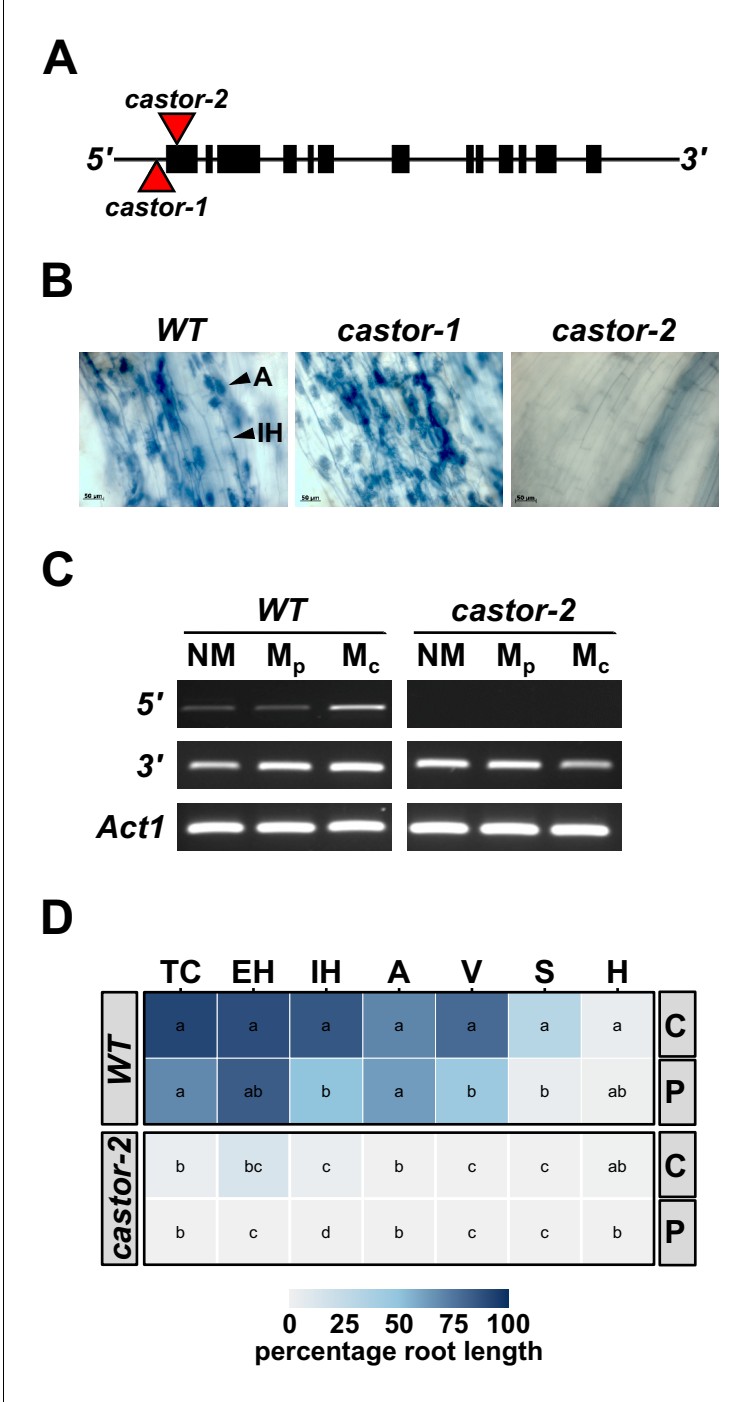

**Figure 1.** Mutation of maize *Castor* results in resistance to AMF. (**A**) Schematic representation of the maize *Castor* gene (GRMZM2G099160_T01 gene model). Boxes indicate coding regions. Red triangles indicate the insertion of *Mutator* transposable elements in the alleles *castor-1* and *castor-2*. (**B**) Root segments stained with trypan blue 7 weeks after inoculation with *Rhizophagus irregularis*. Characteristic root-internal hyphal structures, such as intraradical hyphae (IH) and arbuscules (A) , seen in wild-type (WT) and *castor-1* plants are absent from the *castor-2* mutant. (**C**) RT-PCR analysis of *Castor* transcript accumulation in the roots of WT and *castor-2* plants that were non-inoculated (NM) or inoculated with plate ($M_p$. *In vitro* produced) or crude ($M_c$. Sand-pot produced) inoculum. *Castor* cDNA was amplified using a primer set spanning the *castor-2* mutator insertion site (*5′*) and a second set spanning the *3′*-most intron (*3′*). Primers to the maize actin gene *Act1* were used as a control. (**D**) Colonization by AMF (% root length) of the plants analyzed in **C**. TC, total colonization; EH, external hyphae; IH, internal hyphae;

*Figure 1 continued on next page*

*Figure 1 continued*

A, arbuscules; V, vesicles; S, spores; H, hyphapodia. C, crude inoculum. P, plate inoculum. Lowercase letters indicate significant differences among genotypes and inoculum, determined by the Kruskal-Wallis test and LSD. The online version of this article includes the following figure supplement(s) for figure 1:

**Figure supplement 1.** Generation of the F$_{2:3}$ population.
**Figure supplement 2.** Maize *castor-2* mutants are resistant to colonization by AMF.
**Figure supplement 3.** Evaluation of *AMF-S* (*wt*) and *AMF-R* (*castor*) families in heavily managed (PV) and a rain-fed, medium input (Ameca) fields.

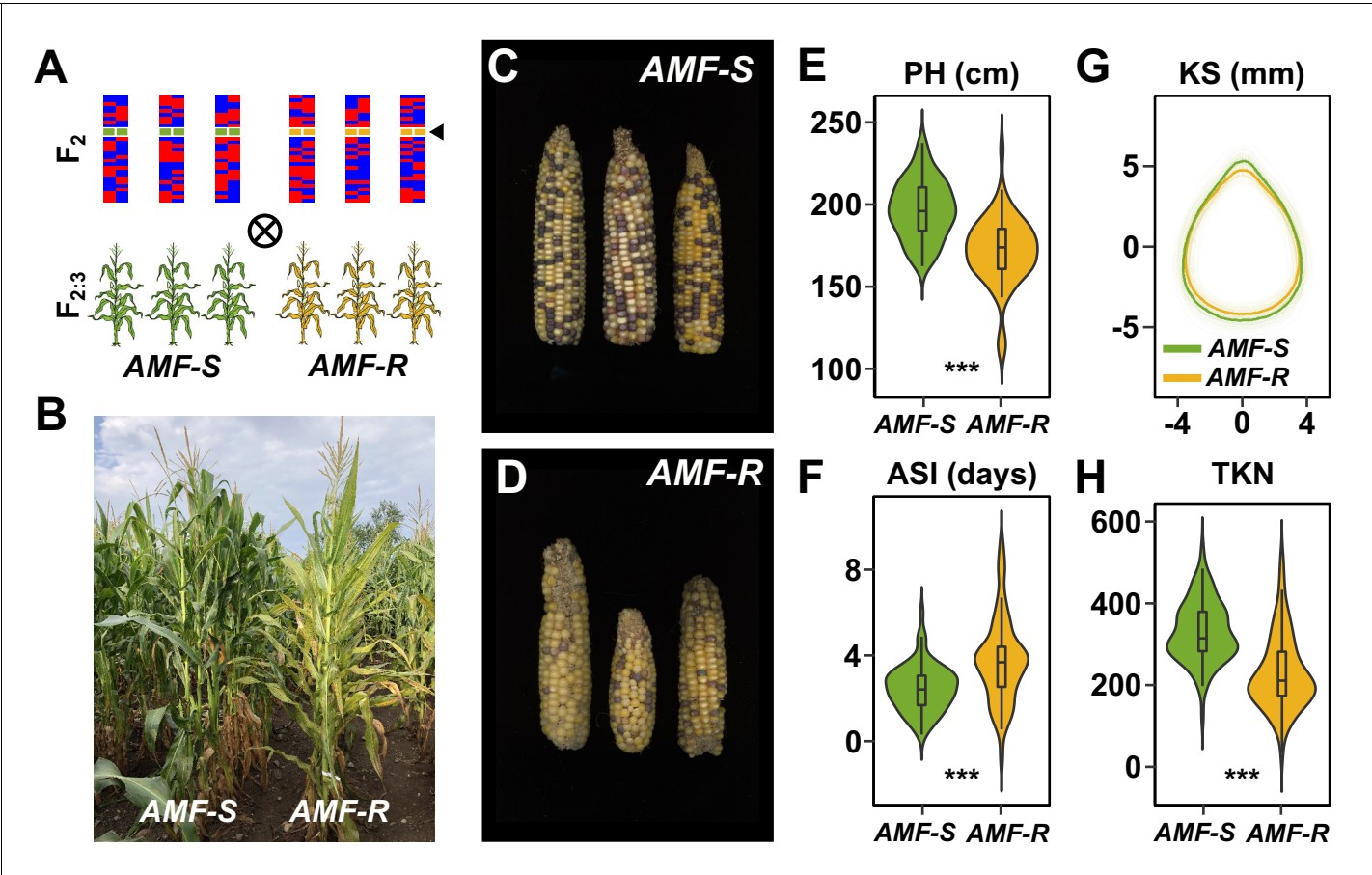

**Figure 2.** Mycorrhizal symbiosis promotes maize growth under cultivation. (**A**) Susceptible (*AMF-S*) and resistant (*AMF-R*) families segregate genomic content from two founder parents, shown as red and blue bars, but are homozygous for the wild-type (green) or mutant (yellow) allele at *Castor* (black arrow), respectively, blocking AM symbiosis in *AMF-R*. (**B**), Border between representative *AMF-S* and *AMF-R* plots, Ameca, Mexico, 2019. (**C**), Representative *AMF-S* ears. (**D**), Representative *AMF-R* ears. (**E**), Plant height (PH) of 73 *AMF-S* (green) and 64 *AMF-R* (yellow) families. (**F**), Anthesis-silking interval (ASI). (**G**), Kernel shape (KS) based on the analysis of scanned kernel images . (**H**), Total kernel number (TKN). The violin plots in (**E, F** and **H**) are a hybrid of boxplot and density plot. The box represents the interquartile range with the horizontal line representing the median and whiskers extending 1.5 times the interquartile range. The shape of the violin plot represents the probability density at different values along the y-axis. ***, difference between *AMF-S* and *AMF-R* significant at p<0.001 (Wilcoxon test; Bonferonni adjustment based on the total trait number).
The online version of this article includes the following figure supplement(s) for figure 2:

**Figure supplement 1.** Location of experimental fields.
**Figure supplement 2.** Representative *AMF-S* and *AMF-R* families in high- and medium- input sites.
**Figure supplement 3.** Plan of the Ameca field design.
**Figure supplement 4.** Comparison of plant phenotypic traits between *AMF-S* (*wt*) and *AMF-R* (*castor*) families.
**Figure supplement 5.** Ear and kernel image analysis.
**Figure supplement 6.** Principal component analysis.

**Table 1.** Summary of the host response and QTLs.

| Trait | Mean AMF-S | Mean AMF-R | Adj. P-value[1] | HR[2] | HR (%)[3] | QTLs[4] | AMFxQTL[5] | Chr[6] | Type[7] |
|---|---|---|---|---|---|---|---|---|---|
| STD | 11 | 10 | NS | 1 | 10 | 0 | 0 | | |
| DTA | 56 | 57 | NS | -1 | -1 | 1 | 1 | 4 | B |
| DTS | 59 | 60 | *** | -1 | -3 | 1 | 1 | 4 | AP |
| ASI | 2 | 4 | *** | -2 | −34 | 1 | 1 | 8 | D |
| PH | 196.8 | 171.8 | *** | 25 | 15 | 2 | 2 | 2, 4 | D, B |
| TBN | 13 | 12 | NS | 1 | 10 | 2 | 1 | 5, 7, | B |
| GC | 0. 56 | 0. 4 | NS | 0.15 | 38 | 1 | 0 | 1 | |
| EW | 86.6 | 60.6 | *** | 26 | 43 | 2 | 2 | 1, 4 | AP, B |
| EL | 13.9 | 12.4 | *** | 1.5 | 13 | 1 | 1 | 1 | D |
| ED | 40.3 | 37.2 | *** | 3.1 | 8 | 2 | 2 | 2, 5 | D, D |
| KRN | 16 | 15 | * | 1 | 6 | 1 | 0 | 2 | |
| KPR | 25 | 18 | *** | 7 | 37 | 2 | 2 | 1, 7 | D, AP |
| KC | 0. 3 | 0.23 | NS | −0.06 | 21 | 1 | 0 | 10 | |
| CD | 26.6 | 26.1 | NS | 0.5 | 2 | 0 | 0 | | |
| FKW | 10.4 | 9.4 | NS | 1 | 10 | 2 | 2 | 1, 10 | D, AP |
| TKW | 67.7 | 44.8 | *** | 22.9 | 51 | 1 | 1 | 1 | AP |
| TKN | 330 | 230 | *** | 100 | 44 | 1 | 1 | 1 | D |
| PC1 | −1.23 | 1.7 | *** | | | 2 | 1 | 1, 5 | D |
| PC2 | 0.126 | −0.182 | NS | | | 1 | 1 | 4 | B |
| PC3 | 0.251 | −0.288 | * | | | 1 | 0 | 7 | |
| PC4 | −0.033 | 0.037 | NS | | | 2 | 1 | 3, 10 | AP |
| PC5 | −0.245 | 0.282 | *** | | | 1 | 1 | 8 | AP |

[1]Wilcoxon tests with Bonferonni adjusted p-values. Note: *: p<0.05; **: p<0.01; ***: p<0.001; NS: not significant. [2]Host response calculated as mean *AMF-S - AMF-R*. [3]Host response expressed as a percentage of *AMF-R* mean. [4]Number of QTL associated with a given trait. [5]Number of QTL showing significant QTL x AMF effect. [6]Chromosomes where the QTLs are located. [7]Pattern of QTL x AMF effect: D: dependence QTL expressed primarily in *AMF-R* families; B, benefit QTL expressed primarily in *AMF-S* families; AP, antagonistic pleiotropy. Trait codes: STD, stand count; DTA, days to anthesis; DTS, days to silking; ASI, anthesis-silking interval; PH, plant height; TBN, tassel branch number; EW, ear weight; EL, ear length; ED, ear diameter; KRN, kernel row number; KPR, kernels per row; CD, cob diameter; FKW, fifty kernel weight; TKW, total kernel weight; TKN, total kernel number. PC, principal components from an analysis of all traits, numbered in descending order of variance explained.

## QTL x AMF effects underlie variation in mycorrhiza response

We performed a Quantitative Trait Locus (QTL) analysis using the trait estimates obtained for $F_{2:3}$ families and the genotypes of their respective $F_2$ parents. We combined *AMF-S* and *AMF-R* families in a single analysis, including the *Castor* genotype as an interactive covariate. Under this model, the *Castor* additive effect (hereafter, AMF effect) estimated the marginal host response across all families, QTL additive effects captured genetic differences between the parents, and QTL × AMF effects indicated underlying heritable differences in the response of CML312 and W22 parents. In addition to directly analyzing trait values, we combined groups of correlated traits by principal component (PC) analysis (*Figure 2—figure supplement 6*), running the first five PCs in QTL mapping. We identified 28 QTLs, of which 21 showed evidence of AMF × QTL interaction (*Figure 3A*, *Figure 3—figure supplements 1* and *2*; *Table 1*). We combined significant QTL, AMF and AMF x QTL terms, into a single multiple-QTL model for each trait and calculated the percentage of phenotypic variation explained by AMF, by additive QTL and by AMF × QTL terms (*Figure 3B*; *Supplementary file 1*). For plant height, ear weight and total kernel weight, the combination of AMF and AMF x QTL effects explained more than half of the total genetic variance (based on an estimation of broad-sense heritability, $H^2$), and over a quarter of the total phenotypic variance (*Figure 3B*). The identification of significant AMF × QTL effects reveals heritable differences in the response to AMF between the CML312 and W22 parents.

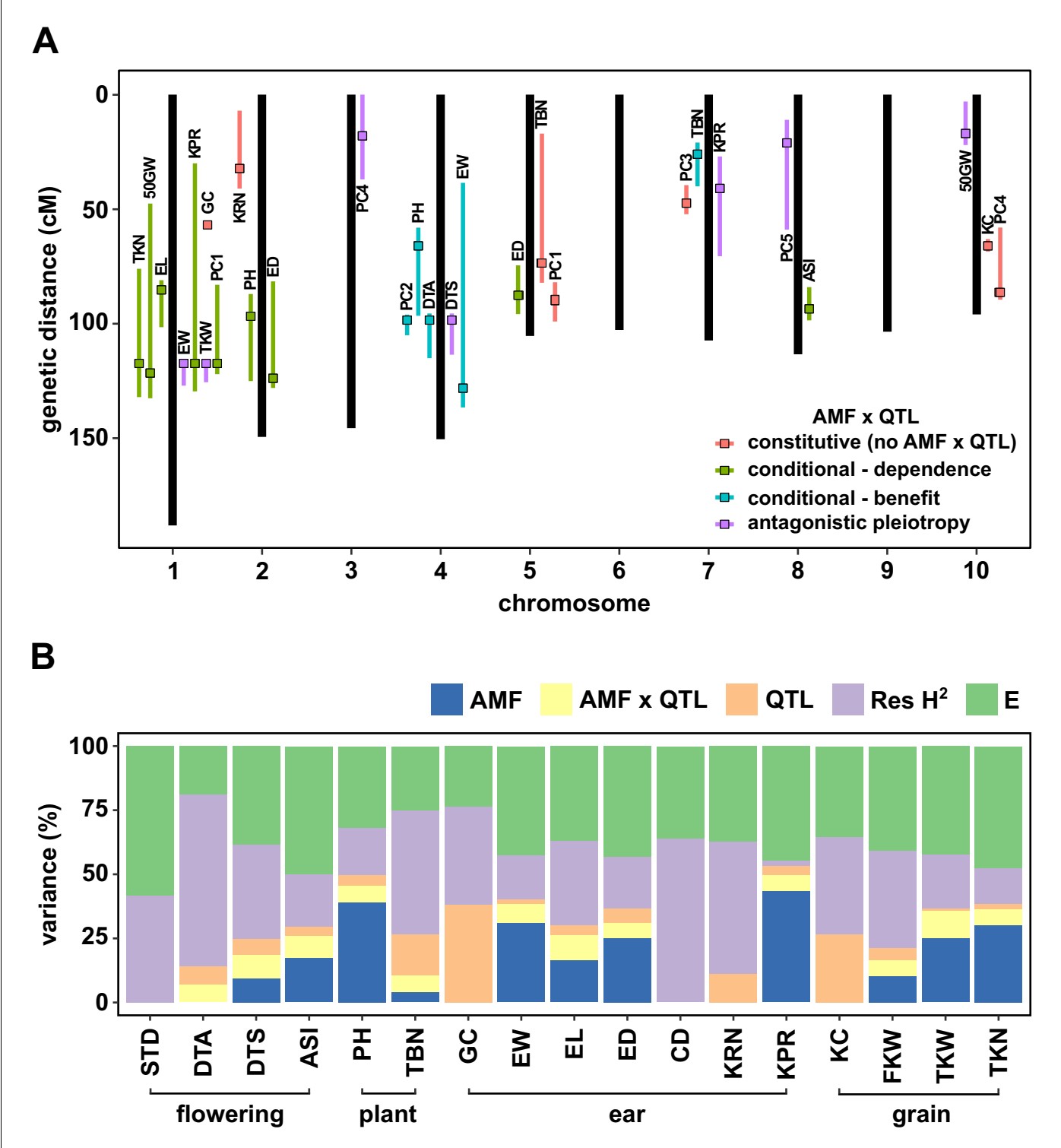

**Figure 3.** QTL x AMF effects contribute significantly to variation in host response. (**A**) Genomic position of QTL identified in this study. Boxes indicate the position of the peak marker. Bars represent the drop-1 LOD interval. Colors denote patterns of AMF × QTL interaction as described in the text. (**B**) The contribution of different components to phenotypic variance in different plant traits among *AMF-S* and *AMF-R* families. Total genetic variance was estimated based on differences between experimental blocks and partitioned into variation explained by the additive effect of AMF, the additional variation explained by interactive QTL and QTL × AMF interaction (QTL × AMF), the additional variation explained by additive QTL (QTL) and residual genetic variation (Res H$^2$). The balance of the phenotypic variance was assigned to the environment (E). Traits codes as in *Table 1*.

The online version of this article includes the following source data and figure supplement(s) for figure 3:

*Figure 3 continued on next page*

*Figure 3 continued*

**Source data 1.** QTL analysis.
**Figure supplement 1.** Genotypic analysis.
**Figure supplement 2.** Genetic map.

## The genetic architecture of response variation distinguishes dependence and benefit

Variation in host response confounds differences in dependence and benefit (*Figure 4A,B*, *Figure 4—figure supplement 1*; *Janos, 2007*; *Sawers et al., 2010*) - the former being the capacity of a given variety to perform in the absence of AMF, the latter the degree to which a plant host profits from the association. Having identified significant AMF × QTL interaction, we distinguished dependence and benefit by inspecting QTL effects separately in *AMF-S* and *AMF-R* families (*Figure 4—source data 1*). For 12 of the 17 QTLs showing significant interaction, the effect was *conditional* - that is expressed predominantly in either the *AMF-S* or *AMF-R* families, but not in both (*Figures 3A* and *4*; *Table 1*; *Des Marais and Juenger, 2010*). We considered conditional QTLs expressed specifically in the *AMF-R* families to represent variation in dependence. Conversely, we considered conditional QTLs expressed specifically in the *AMF-S* families to be variation in benefit (*Figures 3* and *4*; *Table 1*). Two regions of the genome were associated with multiple QTLs, suggesting a shared mechanistic basis: a region on the long arm of chromosome (chr) one linked to ear-traits and a region on chr four linked to plant height and flowering time (*Figure 3A*). Combining traits using PCs further refined these two QTL 'hot-spots' and demonstrated the difference between dependence and benefit (*Figure 4D*. *Table 1*).

## Antagonistic QTL effects suggest a trade-off between mycorrhizal and non-mycorrhizal performance

Several QTLs showed more extreme QTL × AMF in which the effect was expressed in both *AMF-S* and *AMF-R* families, but with a change of sign (*i.e.* a 'swap' in the relative performance of the parental alleles) - a condition described as *antagonistic pleiotropy* (*Figure 4—figure supplements 1*, *Figure 5A*; *Des Marais and Juenger, 2010*). Under this scenario, the superior allele in mycorrhizal plants is detrimental to performance in the absence of mycorrhiza, and *vice versa*, providing evidence for a trade-off at the single-locus level. Significantly, several QTLs associated with key yield components showed antagonistic pleiotropy (*Figure 3*, *Figure 4—source data 1*, *Figure 5B, C*; *Table 1*). We observed analogous 'rank changing' effects at the level of the whole genotype. Although our mapping design did not permit direct evaluation of the same families as both *AMF-S* and *AMF-R*, we could fit QTL models to all genotypes across both levels of AMF. The resulting estimates showed evidence of rank change with respect to performance with or without AMF (*Figure 5D*). To explore further the response at the genotype level, we used whole-genome models to predict mycorrhizal and non-mycorrhizal trait values across the 137 families. Specifically, we used the *AMF-S* families to train a model for mycorrhizal performance that we applied across the whole population. Similarly, we used the *AMF-R* families to train a model for non-mycorrhizal performance. Whole-genome predictions aligned well with the observed values of their training sets, although they did not capture the extreme observed values (*Figure 5E*). Comparison of the two models indicated that genotypes associated with the highest values of the major yield components in one condition were unexceptional or poor in the other (*Figure 5F*), indicating a trade-off between dependence and benefit.

## Discussion

We have presented evidence that AM symbiosis makes a significant contribution to maize performance in a medium-input, rain fed, subtropical field. Extrapolating our observations to a cultivated field planted at 80,000 plants/ha (typical in the locality of the trial site), we estimate that colonization by AMF would contribute ~2 tonnes/ha to a total yield of ~5.5 tonnes/ha. Although a rough figure to be treated with caution, this estimate serves to provide an indication of the importance of AMF, and is consistent with the previous evaluation of an inbred mutant line that linked the loss of

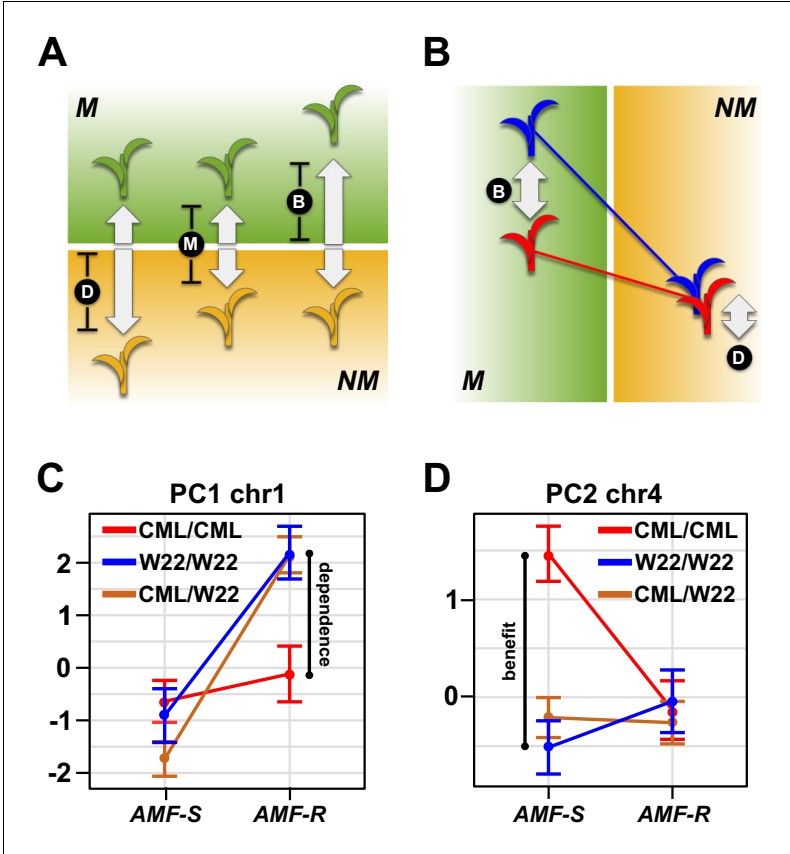

**Figure 4.** Mycorrhiza response confounds benefit and dependence. (**A**) Host response (R) is the difference in trait value between mycorrhizal (M; green) and non-mycorrhizal (NM, yellow) plants. Increased response can result from either greater dependence (D; poorer performance of NM plants) or greater benefit (B; greater performance of M plants). (**B**), QTL × AMF effects underlying variation in response reveal the balance of benefit and dependence. In this theoretical example, the effect is conditional on mycorrhizal colonization, reflecting a difference in benefit more than dependence. (**C, D**) Effect plots for major QTL associated with PC1 and PC2, respectively. Effect of the homozygous CML312 (red), homozygous W22 (blue), or heterozygous (brown) genotypes in *AMF-S* and *AMF-R* families. The PC1 QTL is conditional on *AMF-R*, indicating a difference in dependence. The PC2 QTL is conditional on *AMF-S*, indicating a difference in benefit.

The online version of this article includes the following source data and figure supplement(s) for figure 4:

**Source data 1.** QTL analysis.

**Figure supplement 1.** Scenarios in AMF x genotype interaction.

mycorrhizal phosphate uptake with a 15% reduction in ear-weight (*Willmann et al., 2013*). We have only evaluated one location, collecting data in a single season. It will be important to explore the stability of the mycorrhizal effect in subsequent years and in different locations, under a range of management practices. We can hypothesize that mycorrhiza will be more significant in more stressful environments (*e.g. Sawers et al., 2017*; *O'Brien et al., 2018*). Beyond such broad generalizations, greater quantitative characterization is needed to obtain a more complete picture of the role of AM symbiosis in agriculture , across the full range of agronomic scenarios, from smallholder plot to high-input intensive farm. The approach we describe here can also be used to evaluate the functional impact of practices aimed at promoting or enriching soil health, including the overall 'organic package' or specific interventions such as reduced tillage or application of exogenous AMF or other microbes.

Inbred maize varieties consistently show a positive response to AMF when grown under low to moderate nutrient availability (*e.g. Kaeppler et al., 2000*; *Sawers et al., 2017*). In other crop species, the impact of AMF is more variable, falling along a continuum from mutualism to parasitism (*Johnson et al., 1997*; *Wen et al., 2019*; *Bergmann et al., 2020*). In sorghum, a close relative of

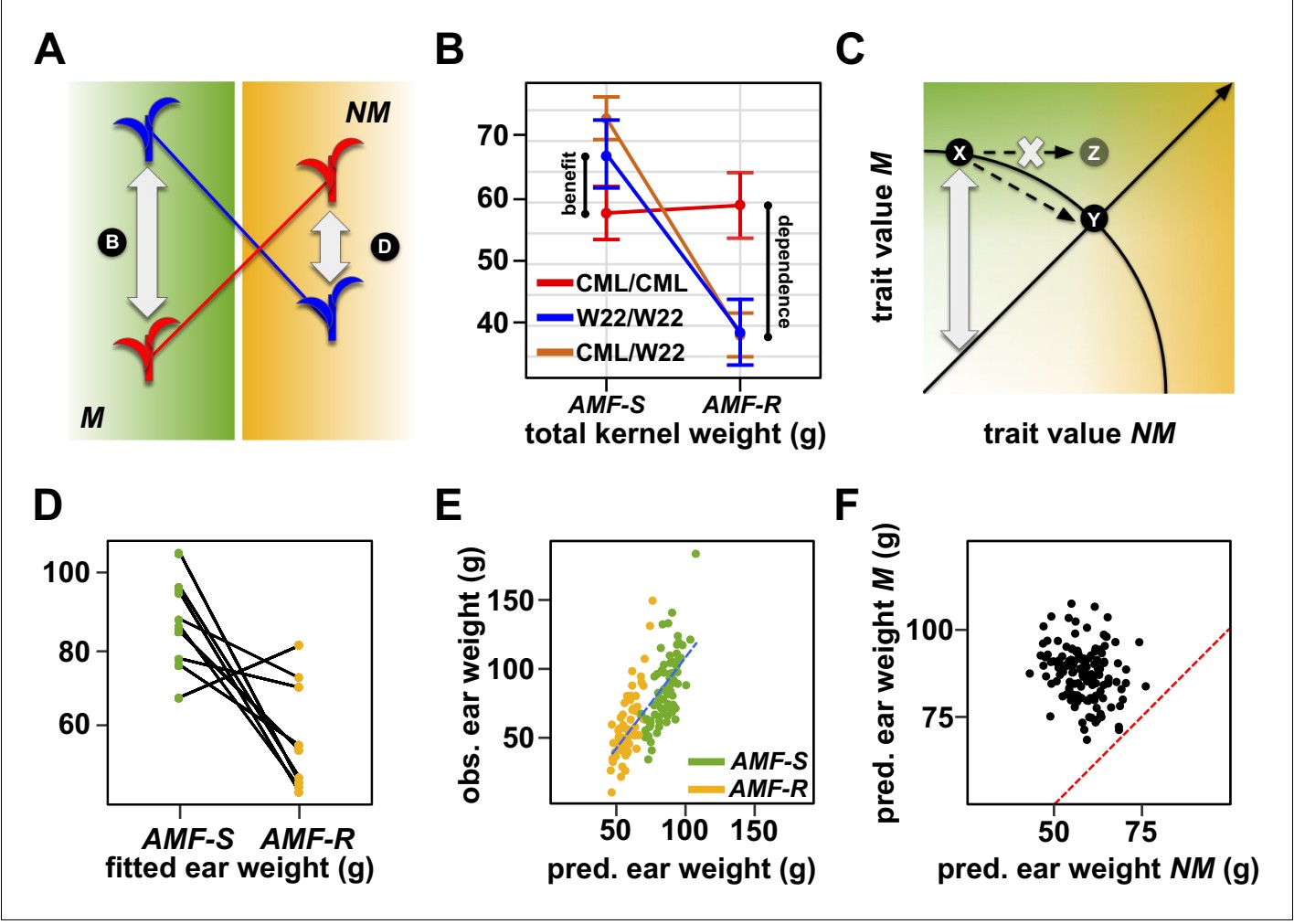

**Figure 5.** The genetic architecture of mycorrhiza response implies a trade-off between dependence and benefit. (**A**) Extreme QTL × AMF effects result in antagonistic pleiotropy. In this theoretical example, the blue allele is superior in mycorrhizal (**M**) plants, while the red allele is superior in non-mycorrhizal (**NM**) plants. Such a QTL is linked to differences in both dependence and benefit. (**B**) Effect plot for a total kernel weight (TKW) QTL located on Chromosome 1. TKW of families homozygous for the CML312 allele at this locus (red) is stable across susceptible (*AMF-S*) and resistant (*AMF-R*) families. In contrast, families homozygous for the W22 allele (blue) or heterozygous at this locus (brown), show superior TKW if *AMF-S* but lower TKW if *AMF-R*. (**C**) Theoretical trade-off between the performance of non-mycorrhizal (NM) and mycorrhizal (M) plants. For any position on this plot, the vertical distance above the diagonal indicates the magnitude of host response. Trade-off prevents occupancy of the upper-right of the plot, defining a so-called 'Pareto front' (solid arc). Here an increase in the NM performance of variety X is necessarily accompanied by a reduction in M performance, as described by movement along the arc to condition Y. Biological constraint prevents the path towards condition Z. (**D**) Fitted ear weight values for all genotypes from a multiple QTL model under both levels of AMF (*AMF-S* and *AMF-R*). Line segments connect the same plant genotype under the two AMF levels. (**E**) Observed ear weight against whole-genome prediction for 73 susceptible (*AMF-S*) and 64 resistant (*AMF-R*) families. Best fit regression line in dashed blue. (**F**) Predicted ear weights for all 137 genotypes based on *AMF-S* (M) and *AMF-R* (NM) whole-genome models. Dashed red line shows *AMF-S* = *AMF*-R, that is, no host response.

maize, host response varies from positive to negative depending on both the plant variety and the AMF species (*Watts-Williams et al., 2019*). However, host response is typically measured in young plants grown under controlled conditions, and the values obtained may under-estimate the true importance of AM symbiosis in the field. In the specific case of maize, it is important to distinguish the inbred lines characterized in most published reports from the hybrids and open-pollinated land-race varieties used in cultivation (*c.f.* other major cereal crops that are naturally inbreeding). Heterosis ('hybrid vigor') will tend to reduce dependence and, consequently, host response to AMF. This does not imply that arbuscular mycorrhizae are unimportant in hybrid maize cultivation, only that proportional responses may differ with the degree of inbreeding. Our $F_{2:3}$ families are segregating

over ~50% of the genome, any given individual retaining ~25% heterozygosity. As such, this material is intermediate between an $F_1$ hybrid and an inbred line. Introgression of the *castor* mutation into different genetic backgrounds will allow the generation of *AMF-R* hybrids and subsequent evaluation of their performance in field trials. More generally, our approach might be applied readily to other crops provided suitable mutants are available.

We found evidence for substantial differences in host response among plant genotypes. In terms of variance explained, AMF $\times$ QTL effects were more important than the QTL main effects for total kernel weight, total kernel number, and fifty-kernel weight. We saw no evidence that our W22 and CML312 parents, as 'modern' inbred lines, had lost the capacity to benefit from AM symbiosis (*e.g.* see *Koide et al., 1988*; *Hetrick et al., 1992*). Furthermore, the identification of QTL linked to benefit suggests that it would be possible to improve the capacity of plants to profit from the symbiosis by combining positive alleles. Additional characterization is required to discover the mechanistic basis of the differences we saw in the host response. Although we have yet to conduct a detailed quantification of colonization across all families, a greater abundance of root internal fungal structures is not necessarily an indicator of greater host response (*Kaeppler et al., 2000*; *Sawers et al., 2017*; *Watts-Williams et al., 2019*; *Plouznikoff et al., 2019*; *Pawlowski et al., 2020*; *Huang et al., 2020*). Indeed, the capacity of the host to limit fungal colonization under certain conditions may maximize plant benefit, and plants will restrict colonization under high nutrient conditions (*Karlo et al., 2020*; *Nouri et al., 2014*; *Sawers et al., 2017*). It has been proposed that the absence of the high-colonization allele of *OsCERK1* from *japonica* rice reflects the action of human selection (*Huang et al., 2020*), consistent with the idea that greater colonization may not always beneficial. In terms of nutrient foraging, the extent of the external hyphal network may be more significant than the level of root colonization (*Yao et al., 2001*; *Munkvold et al., 2004*; *Schnepf et al., 2008*; *Sawers et al., 2017*). Overall, many factors will interact to determine the symbiotic outcome. It has been shown that certain bacterial taxa exert beneficial and synergistic effects on AM symbiosis, acting as so-called 'mycorrhiza helpers' (*Frey-Klett et al., 2007*; *van der Heijden et al., 2016*; *Ferreira et al., 2020*). In *Lotus japonicus*, disruption of the AM symbiosis impacts other members of the root-associated microbial community (*Xue et al., 2019*). Characterization of the root microbiome of *AMF-S* and *AMF-R* families will provide insight into the interplay between AMF, the broader microbial community, and host response.

Our analysis indicated a functional trade-off between host dependence and benefit: the genotypes predicted to be the best in AM susceptible families were predicted to be unremarkable or poor in the context of AM resistance, and *vice versa*. At the level of individual loci, examples of antagonistic pleiotropy (*Des Marais and Juenger, 2010*) were seen for QTL associated with days-to-silking, ear weight, total kernel weight, number of kernels per row, and fifty kernel weight. Our identification of intraspecific genetic trade-offs is analogous to differences seen in plant nutritional strategy at higher taxonomic levels, where the demands placed on plant anatomy and physiology by direct foraging or engagement in AM symbiosis appear to prevent co-optimization for both (*Lambers et al., 2008*; *Wen et al., 2019*). Detailed anatomical characterization has also suggested that such trade-offs will exist. For example, increasing the proportion of root cortical aerenchyma (root air space) reduces the carbon demand of the root system, promoting foraging efficiency (*Postma and Lynch, 2011*), but at the cost of limiting colonization by AMF and, potentially, host response (*Strock et al., 2019*). It was somewhat surprising to see a QTL for a morphological trait such as tassel branch number to be conditional on susceptibility to AMF, indicative of the action of pleiotropic systemic signals. Indeed, the same phytohormones that play a key role in plant development (including patterning of tassel architecture) are also implicated in the establishment and regulation of AM symbiosis (*Gutjahr, 2014*; *Bonfante and Genre, 2015*). For example, the disruption of ethylene signaling has been shown to modulate the level of (negative) host response to AMF in *Nicotiana attenuata* (*Riedel et al., 2008*); a DELLA repressor in the gibberellic acid signaling pathway is essential for colonization by AMF in rice ( *Yu et al., 2014*); a feedback-loop involving strigolactone modulates the extent of colonization by AMF in *Medicago truncatula* (*Müller et al., 2019*). It may be significant that many of these same signals have been the targets of domestication and plant improvement (*Sawers et al., 2018*).

QTL $\times$ AMF effects were in many cases driven by plasticity associated with the temperate W22 allele; that is, effects linked to alleles from the better adapted sub-tropical CML312 parent were more stable with respect to the presence or absence of AMF in our sub-tropical field. This same

result might be interpreted as evidence of the AM symbiosis mitigating or buffering the deleterious effect of 'non-native' W22 alleles. It will be interesting to characterize additional host genotypes, including those considered to be more responsive to AMF (*e.g. Sawers et al., 2017*) and landrace and wild-relative diversity. It has been suggested that mutalisms play a role during adaptation to new or stressful environments (*O'Brien et al., 2018*; *O'Brien et al., 2019*). In addition to affecting yield, the genetic architecture we observed implies a role for AMF in influencing yield stability. Given the unpredictability in the AM community across location or year, optimization of the plant host might require consistency of response as much as maximum benefit. Although we have found no evidence of selective pressures to reduce colonization by AMF in maize breeding, the trade-offs we describe suggest that efforts to optimize plants for better performance either with or without the contribution of microorganisms may not be easily aligned. Ultimately, we may need to make decisions about the types of agroecosystems we want and develop our crops accordingly.

## Materials and methods

### Plant material and generation of the $F_{2:3}$ mapping population

The *castor-1* (*mu1018108*; stock UFMu-01071) and *castor-2* (*mu1045205*; stock UFMu-05472) alleles were identified as insertions in the gene GRMZM2G099160/Zm00001d012863 in the publicly available Uniform Mu collection (www.maizegdb.org; *McCarty et al., 2005*). The *castor-1* insertion was located 44 bp upstream of the translational start site. The *castor-2* event was located 39 bp downstream of the translational start site. The $F_{2:3}$ population was developed from the cross between a stock homozygous for the *castor-2* allele in the W22 background and the subtropical CIMMYT (www.cimmyt.org) inbred line CML312. The initial $F_1$ was self-pollinated to generate an $F_2$ segregating *castor-2* and W22/CML312 genome content. $F_2$ plants were genotyped by PCR to identify homozygous wild-type and mutant individuals, and these self-pollinated to generate susceptible (*AMF-S*) and resistant (*AMF-R*) $F_{2:3}$ families, respectively. $F_{2:3}$ families were increased by sibling-mating within families. Material was generated in Valle de Banderas, Nayarit, Mexico and Irapuato, Guanajuato, Mexico from 2016 to 2019. Small amounts of seed are available for distribution through direct contact with the authors.

### Mycorrhizal colonization assays

Wild-type (W22) and mutant seedlings were inoculated with *Rhizophagus irregularis* - either 5 mL of crude inoculum (produced by sand culture with *Tagetes multiflora*) or 500 fungal spores isolated from transgenic hairy root carrot root cultures. Plants were grown in a growth chamber at 28/20℃, 65% humidity, and 12 h / 12 h day/night cycle. Plants were irrigated with water for the first 2 weeks and subsequently twice a week with Hoagland solution adjusted to 100 µM phosphate. Plants were harvested at 7 weeks and fungal colonization quantified microscopic inspection. Root segments (~1 cm in size) were incubated in 10% KOH for 30 min at 96℃, washed three times in distilled water, incubated in 0.3 M HCl for 30–120 min at room temperature, and then heated at 96℃ for 8 min in a 0.1% w/v trypan blue staining solution in a 2:1:1 mixture of lactic acid: glycerol: distilled water. Samples were de-stained with a 1:1 solution of glycerol and 0.3 M HCl and 10 segments per sample were mounted on a cover slide. Fungal colonization was scored as the presence or absence of specific fungal structures at ten points per root piece (*Gutjahr et al., 2008*).

### RT-PCR analysis

RT-PCR to quantify *Castor* transcripts was performed as previously described (*Gutjahr et al., 2008*) using primer pairs flanking the *castor-2* insertion site (HUN1 5' Fwd: CCC CTC GAC CCC GAC TC; HUN1 5' Rev: AAG AGA AGT TCC TGC GGA GA) or spanning the last intron (HUN1 3' Fwd: GTT CCC CGA GGG ACC TTT TC; HUN1 3' Rev: GCC TCA AGA CGG TAC CCA AT). The maize *Act1* gene was used as a control (ACT1 Fwd: GGT GGC TCT ATT TTG GCT TCT; ACT1 Rev: CGT ACC A TG TCG AAC TTC CC).

### DNA extraction and PCR genotyping of *castor*

DNA extraction was performed as described in *Fulton et al., 1995*. The genotype at the *Castor* locus was determined by running wild-type and mutant PCR reactions. To amplify the wild-type

fragment, a pair of gene-specific primers were used (HUNF01-CGCGAAGAAACGCAGACATTCC and HUNR04-TAACCTGGAGCGAACAGAATCCAC), generating a product of 606 bp. To amplify the mutant fragment a combination of *Mutator* primer and gene-specific primer was used (HUNF03-C TTGGGCGCATTGGAAATTCATCG and RS183-CGCCTCCATTTCGTCGAATCCSCTT), generating a fragment of 839 bp. PCR conditions were: 1 cycle of initial incubation at 94℃ for 3 min, 32 cycles of 94℃ for 30 s, 60℃ for 30 s and 72℃ for 1 min, and 1 cycle of a final extension at 72℃ for 5 min. PCR used the Kapa Taq PCR kit from Kapa Biosystems (Wilmington, MA) following the manufacturer's instructions. Products were visualized on 1% agarose gel.

## Whole-genome genotyping and genetic map construction

Approximately 200 $F_2$ parents were analyzed using the Illumina MaizeLD BeadChip (https://www.illumina.com/products/by-type/microarray-kits/maize-ld.html). Approximately 3000 single nucleotide polymorphisms (SNPs) were detected, with a call rate of ~96% for all samples. The SNP set was processed using TASSEL 5 (*Bradbury et al., 2007*). SNPs were transformed to ABH format (A: CML312; B: W22) and filtered to remove sites matching any of the following criteria: >30% missing data; monomorphic sites outside of the *Castor* locus; SNPs outside the *Castor* locus showing segregation distortion ($X^2$ test, p<0.05); redundant sites. The markers '5_2269274', '5_3096229', and '5_4270584' (physical position on chromosome 5 at 2.67, 3.1, and 4.27 Mb, respectively) were used to confirm the expected genotype at the *Castor* locus. The genetic map was built in R (R core team 2020) using R/ASMAP (*Taylor and Butler, 2017*) and R/qtl (*Broman et al., 2003*; *Broman and Sen, 2009*).

## Field evaluation and phenotypic analysis

The $F_{2:3}$ population was evaluated in the summer of 2019 at the UNISEM experimental station in Ameca, Jalisco, Mexico (20.78,–105.243). Three complete blocks of 73 *AMF-S* and 64 *AMF-R* families were evaluated. Within each block, *AMF-S* and *AMF-R* families were alternated. The order of the families within the *AMF-S* or *AMF-R* sub-populations was randomized within each block. For each replicate, 45 seeds of each family were sown in a plot of three 2 m long rows. Only the second row of each plot was considered for evaluation, the first and third rows providing a buffer between adjacent families. Phenotypic data were collected at flowering and after harvest as follows: stand count (STD), number of plants per row; days to anthesis (DTA), number of days from planting until anthers visible on the main spike of half of the plants in the row; days to silking (DTS), number of days from planting until silks visible on half of the plants in the row; anthesis-silking interval (ASI), difference in days between anthesis and silking; plant height (PH), distance in cm from the ground to the flag leaf; tassel branch number (TBN), number of primary lateral branches originating from the main spike; ear weight (EW), the weight in g of the ear; ear length (EL), distance in centimeters from the base to the tip of the ear; ear diameter (ED), the diameter in cm at the middle part of the ear; kernel row number (KRN), number of rows in the middle part of the ear; kernels per row (KPR), number of kernels in a single row from the base to the tip of the ear; cob diameter (CD), diameter in cm at the middle part of the cob; fifty kernel weight (FKW), weight in g of 50 randomly selected kernels; total kernel weight (TKW), weight in g of all kernels on the ear; total kernel number (TKN), estimated from TKW/FKW * 50.

## Image analysis

An image-based phenotyping method was used to quantify the size and shape of maize ear, cob, and kernel. Ears and cobs (1-3) from each plot, and approximately 50 kernels from associated ears, were scanned using an Epson Perfection V600 scanner at 1200 dots per inch. Scanned images were uploaded to Cyverse (*Merchant et al., 2016*) and analyzed using an automatic pipeline to quantify ear, cob, and kernel attributes (*Miller et al., 2017*). Width profiles of ear and cob and contour data of kernels were extracted in R to compute their shapes.

## Principal component analysis

Principal Component Analysis (PCA) was conducted to analyze major sources of plant phenotypic variance and to visualize the relationships among all measured phenotypic traits across two maize families. PCA was performed using the prcomp function in R.

## QTL mapping and whole-genome prediction

QTL mapping was performed using the R/qtl package (*Broman et al., 2003*) on a population composed of 73 AMF-S and 64 AMF-R F2:3 families. Single-QTL standard interval mapping was run using the scanone function with Haley-Knott regression, with the genotype at *Castor* (wt = 1; mutant = 0) treated as a covariate (hereafter, AMF). To assess QTL × AMF interaction (*Broman and Sen, 2009*), QTL analysis was performed under four different models: separate analyses for AMF-S ($H_{0wt}$) and AMF-R ($H_{0mut}$) subpopulations; considering AMF as an additive covariate ($H_a$) and considering AMF as an interactive covariate ($H_f$). The LOD significance threshold for each trait and model was established with a permutation test (α = 0.05, 1000 permutations). Evidence of QTL × AMF interaction was obtained by comparing $H_f$ and $H_a$ models, where the difference $LOD_i = LOD_f - LODa$ was considered with reference to the threshold difference $LOD\_thr_i = LOD\_thr_f - LOD\_thr_a$ for evidence of possible interaction. Individual QTL were combined into multiple-QTL models on a per trait basis. Where evidence was found of QTL × AMF interaction in the single scan, the interaction was also included in the multiple-QTL model. Multiple-QTL models were evaluated using the fitqtl function and non-significant (α = 0.1) terms removed according to the drop-one table.

To compare the potential of a given genotype in the presence or absence of colonization, we estimated the performance of the 137 families for both levels of AMF using two approaches. In the first approach, we generated fitted values from the multiple-QTL models, under both levels of the AMF covariate. In the second approach, we performed whole-genome prediction using the R/rrBLUP package (*Endelman, 2011*). Missing genotypes were imputed with the A.mat function and marker effects and BLUE values were obtained with the mixed.solve function. The observed values for AMF-S families were used to train a mycorrhizal model that was then applied to the genotypes of the AMF-R families; the observed values from the AMF-R families were used to train a non-mycorrhizal model that was then applied to the genotypes of the AMF-S families. We used predicted values for all genotypes in both levels of AMF, avoiding possible bias in extreme observed values.

## Acknowledgements

We thank Beda Angehrn and Mario Rivera (UNISEM) for assistance with field evaluation, Cruz Robledo (PV Winter Nurseries) for support with generating material, and Jessica Carcaño-Macias for technical support and seed stock management. We acknowledge Karina Picazarri-Delgado for assistance with preliminary mutant characterization, Nathan Miller for supporting the image analysis, and editors and reviewers for valuable feedback. The maize *castor* mutant was first described in the M. Sc. thesis Ramírez-Flores MR. (2015) *Caracterización genética de mutantes de HUN e IXBA, ortólogos de maíz de los canales de potasio CASTOR y POLLUX*. Centro de Investigación y de Estudios Avanzados del Instituto Politécnico Nacional, Mexico. This study was funded by the Mexican Comisión Nacional para el Conocimiento y Uso de la Biodiversidad (CONABIO) project *Impact of native arbuscular mycorrhizal fungi on maize performance* (N° 62, 2016–2018). MRR-F was supported by a Ph.D. scholarship from the Mexican Consejo Nacional de Ciencia y Tecnología (CONACYT). DA and UP are supported by the research project Engineering the Nitrogen Symbiosis for Africa (ENSA), which is funded by a grant to the University of Cambridge by the Bill & Melinda Gates Foundation and the Foreign, Commonwealth & Development Office (FCDO). RJHS is supported by the USDA National Institute of Food and Agriculture and Hatch Appropriations under Project #PEN04734 and Accession #1021929.

## Additional information

### Funding

| Funder | Grant reference number | Author |
|---|---|---|
| La Comisión Nacional para el Conocimiento y Uso de la Biodiversidad (CONABIO), Mexico | Impact of native arbuscular mycorrhizal fungi on maize performance (N° 62, 2016-2018) | Ruairidh JH Sawers |
| Consejo Nacional de Ciencia y Tecnología (CONACYT), Mex- | Ph.D. scholarship | Maria Rosario Ramirez-Flores |

| ico | | |
|---|---|---|
| U.S. Department of Agriculture | Hatch Appropriations under Project #PEN04734 and Accession #1021929 | Ruairidh JH Sawers |
| Bill & Melinda Gates Foundation and the Foreign, Commonwealth & Development Office (FCDO) | Engineering the Nitrogen Symbiosis for Africa (ENSA) | Doris Albinsky Uta Paszkowski |

The funders had no role in study design, data collection and interpretation, or the decision to submit the work for publication.

### Author contributions

M Rosario Ramírez-Flores, Conceptualization, Data curation, Formal analysis, Funding acquisition, Investigation, Methodology, Writing - original draft, Writing - review and editing; Sergio Perez-Limon, Data curation, Formal analysis, Investigation, Visualization, Methodology, Writing - original draft, Writing - review and editing; Meng Li, Data curation, Formal analysis, Visualization, Writing - original draft, Writing - review and editing; Benjamín Barrales-Gamez, Data curation, Investigation, Methodology, Writing - review and editing; Doris Albinsky, Data curation, Formal analysis, Investigation, Methodology, Writing - review and editing; Uta Paszkowski, Víctor Olalde-Portugal, Conceptualization, Resources, Supervision, Funding acquisition, Investigation, Methodology, Writing - review and editing; Ruairidh JH Sawers, Conceptualization, Resources, Data curation, Formal analysis, Supervision, Funding acquisition, Investigation, Visualization, Methodology, Writing - original draft, Project administration, Writing - review and editing

### Author ORCIDs

M Rosario Ramírez-Flores (iD) https://orcid.org/0000-0003-2561-0086
Sergio Perez-Limon (iD) https://orcid.org/0000-0002-1893-4325
Meng Li (iD) https://orcid.org/0000-0002-6411-3085
Benjamín Barrales-Gamez (iD) https://orcid.org/0000-0002-5264-7637
Uta Paszkowski (iD) http://orcid.org/0000-0002-7279-7632
Ruairidh JH Sawers (iD) https://orcid.org/0000-0002-8945-3078

### Decision letter and Author response

Decision letter https://doi.org/10.7554/eLife.61701.sa1
Author response https://doi.org/10.7554/eLife.61701.sa2

# Additional files

### Supplementary files

• Supplementary file 1. Table S1: Traits measured in this study; Table S2: Trait broad-sense heritability and variance explained by terms in QTL models.

• Transparent reporting form

### Data availability

Genotype and Phenotype data are provided on Figshare under the doi's: https://doi.org/10.6084/m9.figshare.12869867 and https://doi.org/10.6084/m9.figshare.12869666 respectively.

The following datasets were generated:

| Author(s) | Year | Dataset title | Dataset URL | Database and Identifier |
|---|---|---|---|---|
| Sergio PL, Ruairidh S, Rosario RFM, BenjaminBG, Meng L | 2020 | CML312 x W22(hun1-2) F2 population genetic map | https://doi.org/10.6084/m9.figshare.12869867 | figshare, 10.6084/m9.figshare.12869867 |
| Meng L, Ruairidh S, | 2020 | AMECA_2019_phenotypic_dataset. | https://doi.org/10.6084/ | figshare, 10.6084/m9. |

Sergio PL,  Rosario RFM,  Gamez B　　　　xlsx　　　　m9.figshare.12869666　　　figshare.12869666

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
