## [Decision Letter]

**Acceptance summary:**

The interaction of plants and fungi that benefits plant growth is foundational to both wild and agricultural ecosystems. However, the genetic basis of how the plant host responds to the arbuscular mycorrhizae has not been readily amenable to dissecting. This manuscript utilizes a blend of quantitative and mendelian genetics to begin investigating the genetic basis of how arbuscular mycorrhizae influence plant growth. In this work, they begin to unveil a complex genetic underpinning to this interaction.

**Decision letter after peer review:**

Thank you for submitting your article "The genetic architecture of host response reveals the importance of arbuscular mycorrhizae to maize cultivation" for consideration by *eLife*. Your article has been reviewed by three peer reviewers, one of whom is a member of our Board of Reviewing Editors, and the evaluation has been overseen by a Reviewing Editor and Meredith Schuman as the Senior Editor. The reviewers have opted to remain anonymous.

The reviewers have discussed the reviews with one another and the Reviewing Editor has drafted this decision to help you prepare a revised submission.

Summary:

In this work, the authors blend Mendelian and quantitative genetics to begin investigating how AMF influence plant performance in the field.

Essential revisions:

The key revisions are several fold. Typically, QTL studies are replicated either in year, chamber or environment. Given that this is a preliminary report providing an illustration of how to address the genetics of AMF-plant performance links, the evaluators settled on a compromise. The limitations of using a single environment/year in quantitative genetics should be discussed. This should include how these limitations might influence the conclusions. Additionally, it was judged that the manuscript would be strengthened by having a forward-looking section of the Discussion section that would provide ideas and thoughts on what hypotheses raised by this work are most interesting to study and how they might be studied. Finally, there were a number of suggestions and requests made to improve the readability, as this paper has significant potential, it would be great to take this time to maximize its readability to the general audience. For example, this manuscript will pull in specialists from many fields that are maybe not familiar with terms from other fields like the quantitative genetics and any effort to simplify the communication of these concepts will be greatly helpful.

---

## [Author Response]

Summary:In this work, the authors blend Mendelian and quantitative genetics to begin investigating how AMF influence plant performance in the field.Essential revisions:The key revisions are several fold. Typically, QTL studies are replicated either in year, chamber or environment. Given that this is a preliminary report providing an illustration of how to address the genetics of AMF-plant performance links, the evaluators settled on a compromise. The limitations of using a single environment/year in quantitative genetics should be discussed. This should include how these limitations might influence the conclusions. Additionally, it was judged that the manuscript would be strengthened by having a forward-looking section of the Discussion section that would provide ideas and thoughts on what hypotheses raised by this work are most interesting to study and how they might be studied. Finally, there were a number of suggestions and requests made to improve the readability, as this paper has significant potential, it would be great to take this time to maximize its readability to the general audience. For example, this manuscript will pull in specialists from many fields that are maybe not familiar with terms from other fields like the quantitative genetics and any effort to simplify the communication of these concepts will be greatly helpful.

With regard to the Essential revisions, we have reworked the text focusing on the following:

- Caveats – We have been careful to state clearly that our work represents a single evaluation in a single location (*e.g.* Discussion section). We make a number of references to the context specific nature of our results and, indeed, any measure of mycorrhizal response.

- Readability – We have adopted a more narrative style throughout, extending the description of our experimental design and analysis in the main text. Technical terms are defined.

- Perspectives – We have added to Introduction and Discussion section, commenting on further work that might be carried out using the material we describe as well as future developments.